# CollabVR: VR Testing for Increasing Social Interaction between College Students

Diego Johnson, Brayan Mamani and Cesar Salas *

College of Software Engineering, Peruvian University of Applied Sciences, Lima 15023, Peru;
u201714835@upc.edu.pe (D.J.); u20161c629@upc.edu.pe (B.M.)
* Correspondence: cesar.salas@upc.edu.pe

**Abstract:** The impact of the COVID-19 pandemic on education has accelerated the shift in learning paradigms toward synchronous and asynchronous online approaches, significantly reducing students' social interactions. This study introduces CollabVR, as a social virtual reality (SVR) platform designed to improve social interaction among remote university students through extracurricular activities (ECAs). Leveraging technologies such as Unity3D for the development of the SVR environment, Photon Unity Networking for real-time participant connection, Oculus Quest 2 for immersive virtual reality experience, and AWS for efficient and scalable system performance, it aims to mitigate this social interaction deficit. The platform was tested using the sociability scale of Kreijns et al., comparing it with traditional online platforms. Results from a focus group in Lima, Peru, with students participating in online ECAs, demonstrated that CollabVR significantly improved participants perceived social interaction, with a mean of $4.65 \pm 0.49$ compared to traditional platforms with a mean of $2.35 \pm 0.75$, fostering a sense of community and improving communication. The study highlights the potential of CollabVR as a powerful tool to overcome socialization challenges in virtual learning environments, suggesting a more immersive and engaging approach to distance education.

**Keywords:** social virtual reality; virtual learning environments; social interaction; college students; remote education; extracurricular activities; higher education

## 1. Introduction

The COVID-19 pandemic has triggered an unprecedented transformation in education, accelerating the adoption of online approaches, ensuring continuity but leading to decreased social interaction [1,2]. It is important to acknowledge that distance education has been integral to education for over three decades. The pandemic has not initiated but rather expedited the digital transformation, exacerbating challenges in online learning such as loneliness and reduced socialization, both pre-existing concerns [3,4].

Social interaction's significance in the learning and human development process is underscored by UNESCO [5]. Research shows that 71% of students have suffered adverse effects on learning due to a lack of interaction [6]. This isolation leads to a loss of motivation, decline of academic performance and negatively affects educational participation [7,8].

Additionally, there is a growing acknowledgment of the importance of extracurricular activities (ECA) in higher education, particularly in developing countries. Beyond academics, ECAs significantly impact students' personal, professional, and psychological development, along with community participation, fostering motivation, shared knowledge construction, and personal/social skill development [9,10]. Moreover, they cultivate positive attitudes and contribute to a more sustainable society. The increasing interest in ECAs in higher education underscores the need to design remote learning platforms that enhance social interactions among students, a crucial aspect often overlooked in remote classroom environments.

Virtual reality (VR) provides an artificial experience immersing users in a 3D space visually isolated from the physical world [11]. This technology extends to social virtual

reality (SVR), enabling communication through full-body avatars tracked in real time. SVR facilitates interactions that closely resemble face-to-face communication, encompassing elements such as voice, gestures, proxemics, gaze, and facial expressions, demonstrating potential in improving communication and collaboration [12,13]. Despite existing solutions in education with VR technologies, study highlights a lack of the understanding of the unique possibilities of SVR in remote learning experiences, requiring additional practical and empirical evaluations to develop best practices and guidelines for their smooth integration in existing curricula [14].

This study aims to test whether the use of a SVR platform can enhance perceived social interactions among online students. The hypothesis is that students engaging in ECAs through a SVR platform would report an increased perceived social interaction compared to traditional online platforms.

To this end, this article proposes the implementation of CollabVR 1.0.0, a SVR platform designed to enhance social interaction in remote educational contexts through activities such as clubs, presentations, cultural events, and workshops. This proposal addresses the decreased social interaction among students and aligns with the growing recognition of ECAs to improve social engagement and mitigate the adverse effects of remote education.

CollabVR is supported by technologies such as microservices, Unity3D, Photon Unity Networking (PUN), Oculus Quest 2, and AWS. Microservices enable modularization and efficient platform functionalities management, while Unity3D and PUN support the development of an interactive multiplayer virtual environment offering immersive experiences and compatibility with devices like Oculus Quest 2 [15,16]. Finally, Amazon Web Services guarantees scalability and performance by hosting and managing the platform infrastructure.

To assess the perceived level of social interaction within CollabVR, this article will employ the sociability scale developed by Kreijns et al. and validated by Sjølie and van Petegem [17,18]. The experimental approach involves simulating a remote ECA in CollabVR and comparing results with traditional remote platforms.

## 2. Literature Background

Within the context of distance education, the state of the art regarding social interaction in virtual environments has experienced significant growth in recent years. In this section, we will review some works related to social interaction in virtual environments.

Recent research highlights the potential of virtual reality platforms to enhance social interaction within educational environments. For example, Xu [19] demonstrated the benefits of a VR platform designed for a virtual graduation ceremony amid the pandemic, noting improved social interactions and overall satisfaction among participants. Reinforcing this point, comments from several studies highlight the critical role of informal and spontaneous interactions. These interactions not only enhance the learning experience but also play a crucial role in nurturing a sense of community and fostering meaningful connections in remote social environments [20,21].

Beyond social interaction, virtual reality has demonstrated a positive impact on users' psychological well-being. Barreda-Ángeles and Hartmann [22] explored associations between spatial and social presence in VR and perceived psychological benefits. Their findings suggest that greater presence in these environments is linked to more intense feelings of socialization. In a similar study, Siani and Marley [23] found that the recreational use of VR can be beneficial for physical and mental well-being, especially during periods of social isolation.

Communication and interpersonal skills have also benefited from the use of VR. Baccon et al. [24] conducted a comparative study between face-to-face communication, virtual reality, and text-based communication. Their results suggest that VR may be as effective as in-person communication in terms of self-disclosure and interpersonal communication. Yan and Lv [25] supported this idea, stating that communication through VR is more efficient and natural than text-based communication.

Research on the effectiveness of remote teaching has delved into the potential of virtual reality, encompassing non-immersive, semi-immersive, and particularly immersive VR modalities that use head-mounted displays (HMDs) [26], for example. Their immersive nature, HMDs offer unparalleled engagement, but the associated costs and potential discomfort may hinder their widespread adoption among students [27], having conducted comparative analyzes between VR HMD and desktop VR, revealing that the psychological benefit of HMDs users remains consistent across both platforms, aside from the different sense of immersion, there was no significant difference in the overall experience between the two modalities [28].

Freeman and Maloney [12] delved into the realm of SVR, investigating how users present and perceive their identity in these environments. While some users maintain an accurate representation of themselves, many use VR to explore and experiment with different aspects of their identity.

Young et al. [14] emphasize the need for deeper reflection on the implementation of pedagogical methods in virtual reality. The existing literature clearly demonstrates the potential of VR to improve social interaction, psychological well-being, and communication skills in higher education.

## 3. CollabVR Solution

CollabVR, a platform developed by the authors, consists of a VR application and a Web application that are connected using a RESTful API system. This system consists of six microservices, which communicate with each other through an event bus and expose their services through an API gateway. Microservices enable efficient modularity and deployment, facilitate the use of various technologies according to the specific needs of each component, and improve error management [29], key aspects for the effective integration of VR and web applications. Likewise, for real-time connection and collaboration in VR environments, Photon Unity Networking servers are used (Figure 1).

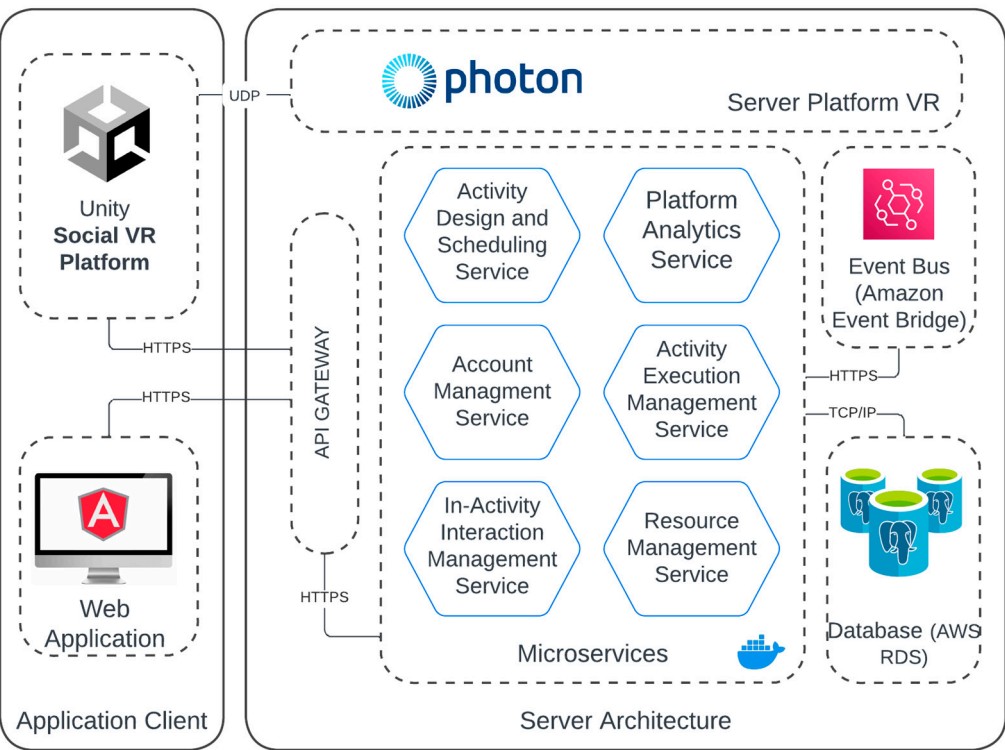

**Figure 1.** Context diagram CollabVR.

The solution is a system that allows universities to manage ECAs and generate greater socialization opportunities for students enrolled remotely, for which CollabVR offers the following functionalities (Figure 2).

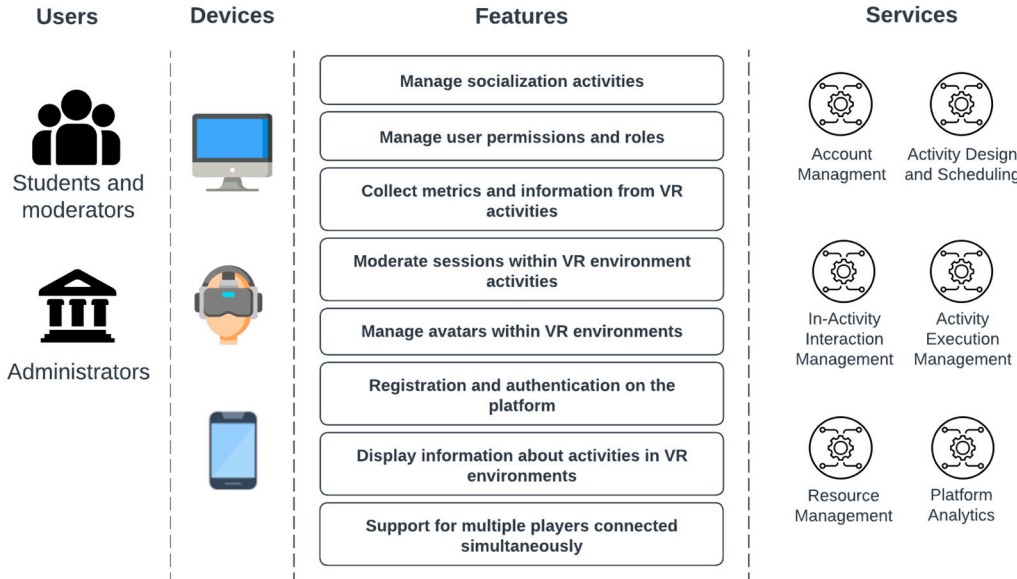

**Figure 2.** CollabVR features and supported devices.

### 3.1. VR Application

CollabVR software is compatible on desktop platforms and with all virtual reality headsets supported by Unity XR Toolkits, for example, the Oculus Quest series. This selection is primarily to provide accessibility to those users who do not have an HMD device.

CollabVR includes four virtual reality environments that have been designed/selected to foster spaces for socialization (Figure 3).

- Open Field: This environment gives users the feeling of being outdoors in nature, encouraging socialization in a natural and relaxed environment.
- Presentation Room: Designed for presentations and group talks, this room facilitates interaction between a presenter and several users, promoting learning and collaboration.
- Night Fire Pit: Provides a quiet and relaxing space for users to gather and socialize in a warm environment, promoting personal connections in an intimate setting.
- Futuristic Room: Offers diversified interaction options with corridors and different rooms, allowing users to have greater control and privacy, adapting to their preferences and needs.

To promote healthy socialization spaces, CollabVR has included moderation functionalities that allow users, in the role of moderator, to regulate the audio of participants and expel them in extreme situations. In addition to providing certain permissions to users within virtual reality rooms, such as the permission to interact with certain objects within the rooms.

Additionally, to facilitate universities in assessing activities, as well as enhancing the overall user experience, CollabVR implements the capability to send metrics of the conducted activities. This includes tracking the number of users connected to activities, identifying real-time activities with the highest user engagement, and monitoring interaction times within the rooms, among other metrics.

Furthermore, given that communication and user identification is essential in the environments, the possibility of customizing the avatar that best suits the user's preferences has been included (Figure 4).

Likewise, within virtual reality environments the user who is speaking can be explicitly visualized. To enhance the overall experience, a microphone audio level modulator and proximity voice chat feature have been incorporated (Figure 5).

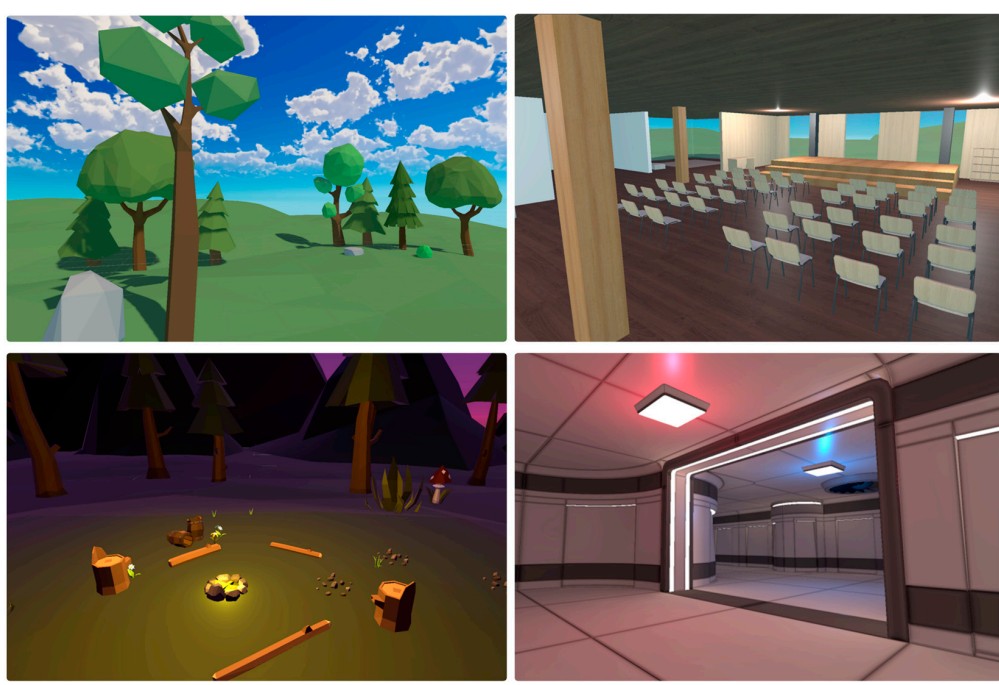

**Figure 3.** Environments intended for socialization included in CollabVR.

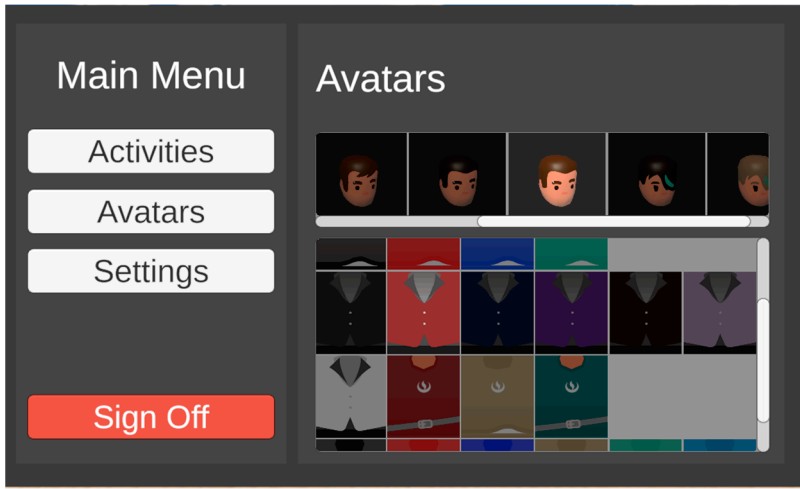

**Figure 4.** Avatar selection menu in CollabVR.

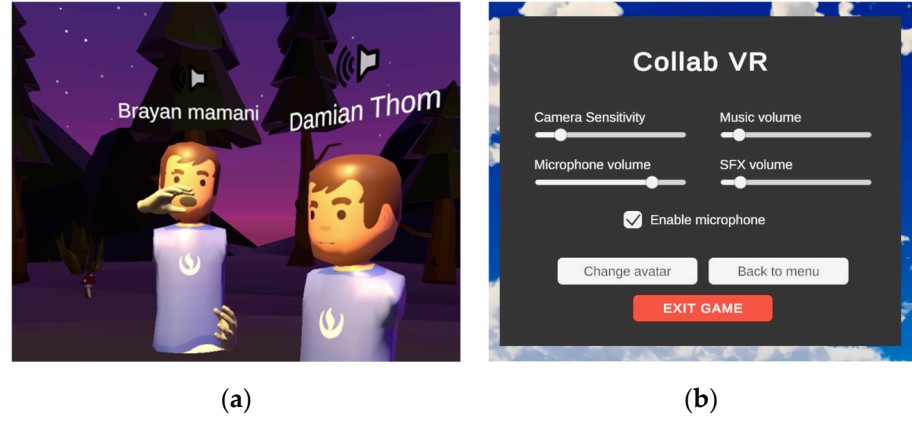

(**a**)                              (**b**)

**Figure 5.** Voice display and modification: (**a**) voice indicator of the participant who is speaking; (**b**) voice/microphone volume control.

### 3.2. Web Application

To allow universities to manage activities more easily, a web application has been developed, enabling users to effortlessly view, create, and assess activity metrics.

Within the activity creation section, a user-friendly form has been designed with three straightforward steps: creation of general details of an activity, schedule and participants of the activity, and selection of the virtual reality environment most in line with the theme of the activity (Figure 6).

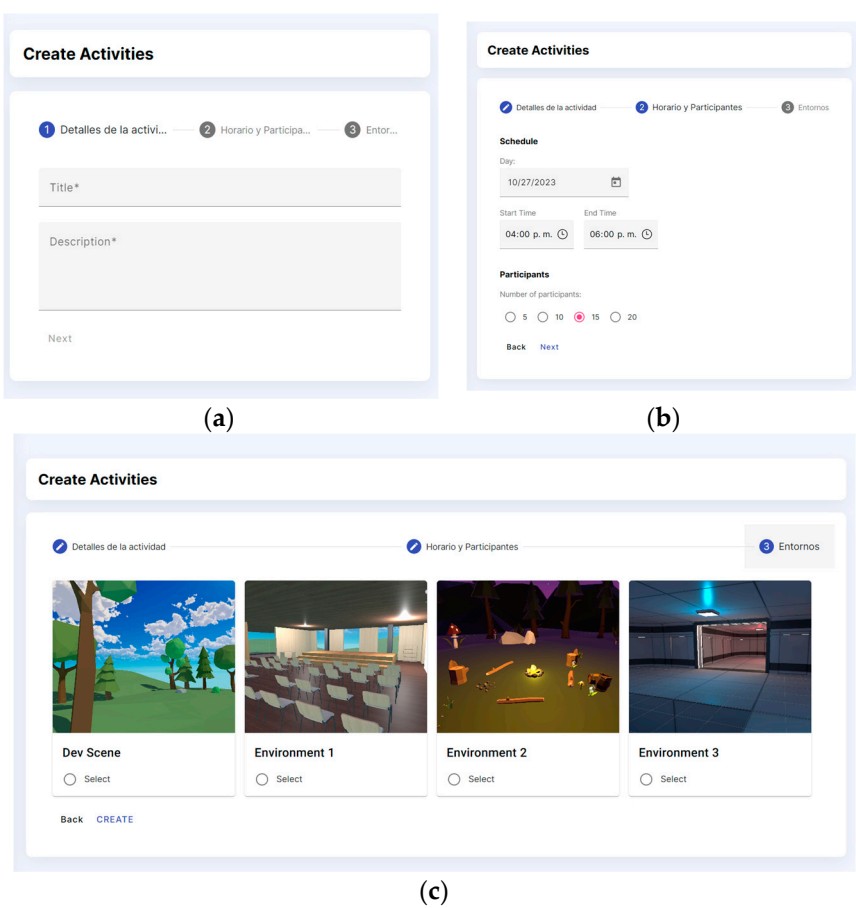

**Figure 6.** Creating activities in the web application. (**a**) Activity detail form; (**b**) schedule and participants form for an activity; (**c**) selection of environments for the VR activity. * means the field is required.

Regarding the registration of users for activities, they can access a list of activities and view the details of a specific activity (refer to Figure 7). In the latter, users can also see the participants registered for the selected activity. In both scenarios, users interested in registering for an activity can do so by utilizing the "Register" button.

Finally, through a dashboard, administrators will be able to view the metrics of the previously conducted activities (Figure 8).

The application is responsible for collecting metrics within the development of ECAs. This includes data such as the time during which a user is participating, the connection time of the users and the number of users within the activities. With these data, information can be extracted, such as the average time of student participation, the level of retention of the activities, and the like. These metrics are essential to provide universities with information about activities development and student engagement. This aims to enable universities to make informed decisions, enhancing their services and adapt to the needs of students in the development of ECAs. While the current set of available metrics is limited, future versions

of the application will incorporate additional metrics, expanding analytical capabilities, and offering a more comprehensive perspective on activities development.

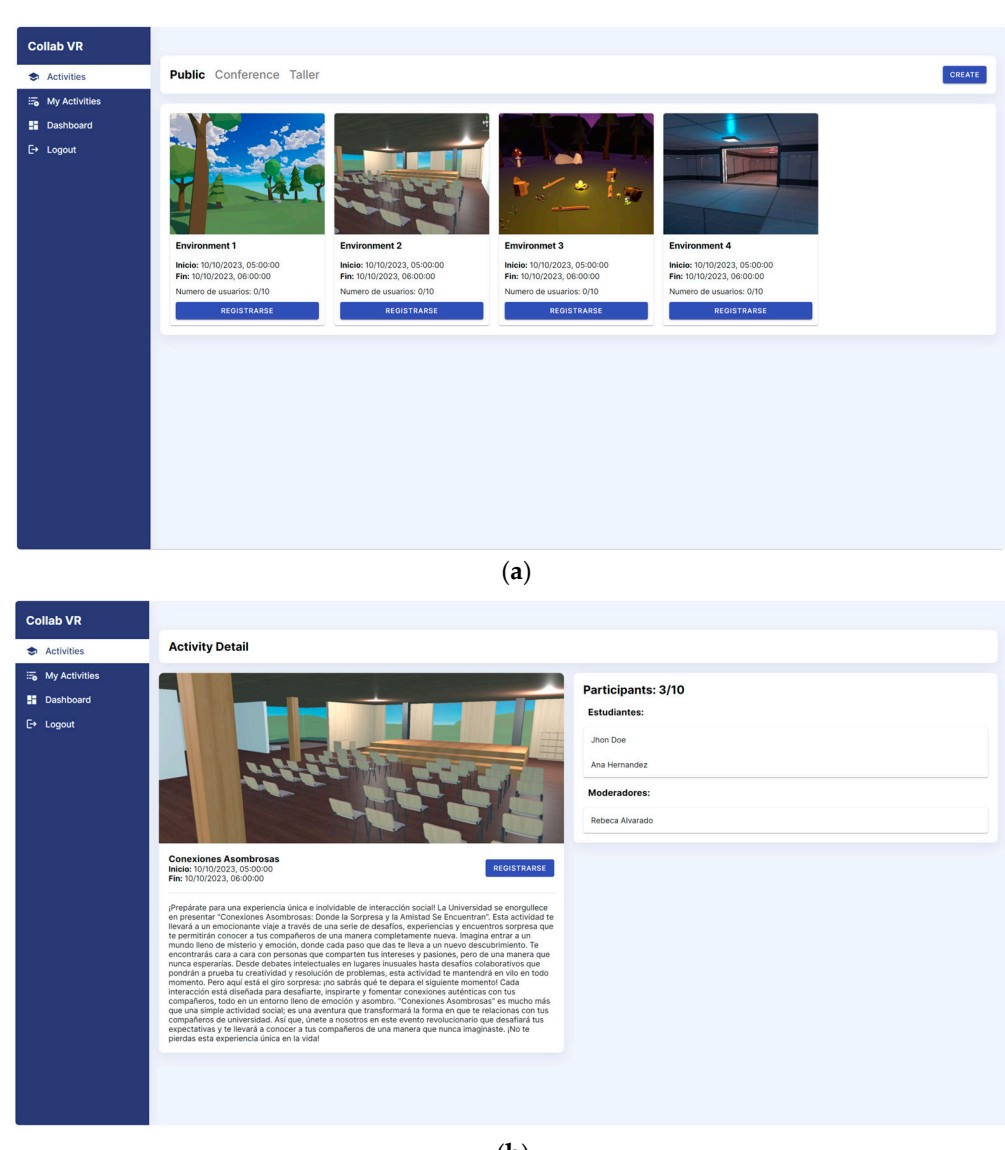

(**a**)

(**b**)

**Figure 7.** Viewing activities and registration to activities: (**a**) list of activities created on the platform; (**b**) detail of selected activity and registered users.

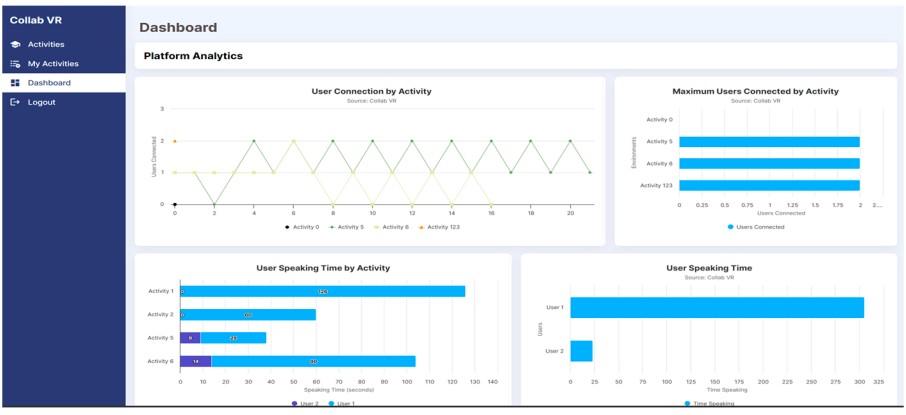

**Figure 8.** Viewing activity metrics.

## 4. Study Design

At the end of the development, we aimed to validate the hypothesis: "Implementing CollabVR for ECAs could increase students' perceived social interaction compared to traditional online platforms". For the planning and development of this experiment, an experiment-driven product development methodology was adopted [30].

To this end, a focus group was organized in the city of Lima, Peru, in October 2023, involving students from the Peruvian University of Applied Sciences (UPC) who had participated in some online ECAs during the 2023-1 and 2023-2 semester. The study included young students aged 18–23 of both genders, engaged in university ECAs, divided into two groups (AS-IS and TO-BE), each consisting of 12 participants. It is worth noting that all participants had prior experience with traditional online platforms and some with virtual reality tools. During the session a questionnaire was administered to each group to evaluate the perception of traditional platforms in contrast to CollabVR.

The questionnaire design is based on the sociability scale by Kreijns et al., which measures the social and emotional aspects of perceived sociability in computer-assisted collaborative learning environments [17]. This scale was chosen for its comprehensive approach to assessing the quality of social space and group dynamics in virtual learning contexts. Its development, grounded in computer-supported cooperative work and human–computer interaction studies, focuses on key elements like group awareness, communication, and community facilitation. Consisting of 10 items, the scale invites participants to express their perceptions on a 5-point Likert-type scale [31]. In addition, open questions were included in the questionnaire with the purpose of collecting feedback on the environment used for online ECAs.

### 4.1. Group AS-IS

This group represented the current situation, in which the students completed the questionnaire based on their previous experiences with traditional online platforms used for online ECAs, with the Blackboard Collaborate tool as the platform for the development of this activities. Students who had previously had experience in developing virtual ECAs taught by the university were recruited through university forums and email.

### 4.2. Group TO-BE

Students in this group, before taking the questionnaire, participated in a one-hour ECA conducted in CollabVR. In this way, their responses reflected their perception of sociability after experiencing the immersive interaction in VR.

For the activity in which they participated, we sought to simulate the "Virtual Public Speaking" workshop offered by the UPC. To achieve this, the "Night Fireplace" environment was chosen to provide a calm and relaxing atmosphere for the 12 students and 2 moderators/instructors involved. The activity was structured in the following stages: We began with an "Introduction and Adaptation to the VR Environment" (10 min) to familiarize students with the application. Moderators offered instructions on navigation and tool use. Next, in "Fundamentals of Public Speaking" (10 min), basic concepts about the structure of speech were taught. The "Practical Activity—Lightning Speeches" (20 min) allowed students to practice short speeches. Students, in their groups, prepared speeches on assigned topics. Each group had 8 min to prepare and 12 min for presentations. During presentations, groupmates provided constructive feedback, reinforcing collaboration and mutual support. To manage anxiety, a section on "Stress and Anxiety Management" (5 min) was included. Facilitators guided students through breathing techniques. This activity was carried out in groups, allowing students to support and reassure each other. Then, "Improvisation and Reaction Exercises" (8 min) were carried out to encourage adaptability. Groups faced improvisation challenges, encouraging adaptability and teamwork when responding to unexpected topics. The activity culminated with a "Question and Answer Session" and "Conclusion and Closing" (both in total 7 min). Students had the opportunity to ask questions and share reflections, both individually and in groups. Finally, the instruc-

tors summarized the learnings and emphasized the importance of continuous practice in public speaking and teamwork.

The objective of this activity was to evaluate how CollabVR is able to improve social interaction and communication skills in a virtual educational context, focusing on teamwork, socialization, and the development of public speaking skills. The activity environment together with the participants are shown in (Figure 9).

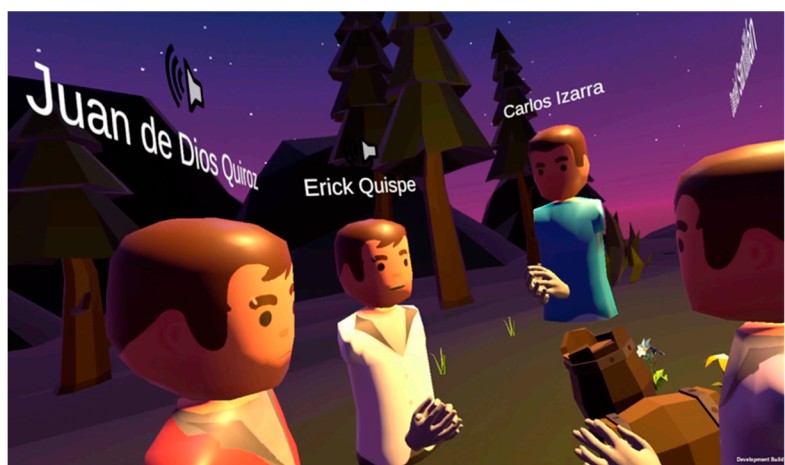

(**a**)

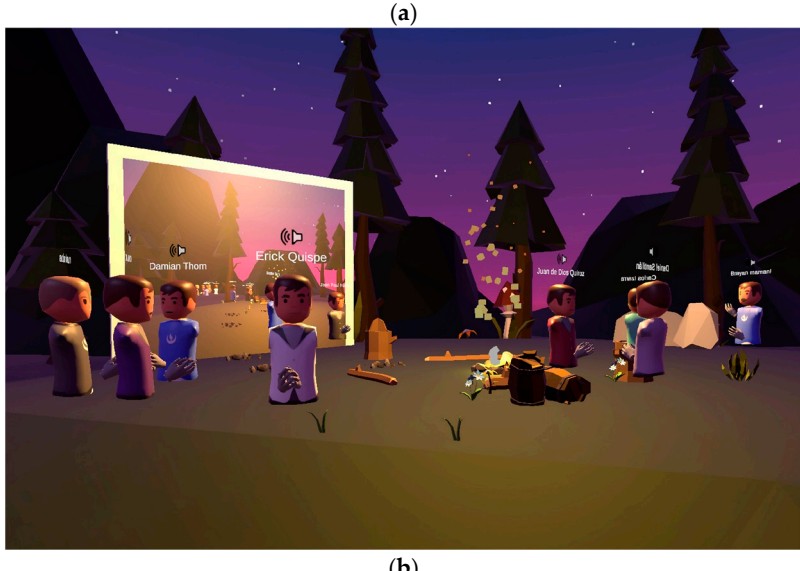

(**b**)

**Figure 9.** Screenshots of the development of the virtual public speaking activity provided through CollabVR: (**a**) capture of the interaction of the groups in the development of the activity; (**b**) capture of the formation of groups for the development of the activity carried out in CollabVR.

## 5. Results

The questionnaires administered after the activity to the "AS-IS" and "TO-BE" groups revealed valuable information about how CollabVR improves perceived social interaction in distance collaborative learning environments compared to traditional platforms such as "Blackboard Collaborate".

Descriptive statistics for each item on the sociability scale are presented in Table 1, and additional detailed information can be found in Appendix A. The mean score on the sociability scale was 2.55 ± 1.37 for the AS-IS group and 4.65 ± 0.49 for the TO-BE group. Within the AS-IS group, the mean scores ranged between 1.92 in the item "This platform allows spontaneous informal conversations" and 3.17 in "I felt comfortable with this platform", the latter value being probably high due to the familiarity of students with

the platform. Meanwhile, the mean scores for the TO-BE group varied between 4.50 and 4.92. In this group, all standard deviations were less than 1, indicating consistency in the perception of sociability with CollabVR. In contrast, in the AS-IS group, 90% of the items had a standard deviation greater than 1, reflecting significant variability in the perception of sociability on traditional platforms.

**Table 1.** Sociability scale items and means for ECAs using traditional online platforms and CollabVR.

| Code | Item | AS-IS | | TO-BE | |
|------|------|-------|------|-------|------|
| | | **M** | **S.D.** | **M** | **S.D.** |
| Q03 | CollabVR allowed me to easily contact my teammates | 2.67 | 1.23 | 4.83 | 0.39 |
| Q04 | I did not feel alone on this platform | 3.17 | 0.94 | 4.83 | 0.39 |
| Q05 | CollabVR allowed me to get a good impression of my teammates | 2.92 | 1.44 | 4.67 | 0.49 |
| Q06 | CollabVR allowed for spontaneous informal conversations | 1.92 | 1.31 | 4.92 | 0.29 |
| Q07 | This platform allowed us to develop as a well-performing team | 2.50 | 1.31 | 4.50 | 0.67 |
| Q08 | This platform allowed me to develop good working relationships with my teammates | 2.33 | 1.56 | 4.50 | 0.80 |
| Q09 | This platform allowed me to identify with the team | 2.33 | 1.56 | 4.58 | 0.51 |
| Q10 | I felt comfortable with this platform | 3.17 | 1.47 | 4.67 | 0.49 |
| Q11 | This platform allowed for conversations unrelated to the workshop task | 2.25 | 1.48 | 4.83 | 0.39 |
| Q12 | This platform allowed me to establish close friendships with my teammates | 2.25 | 1.42 | 4.67 | 0.49 |
| **Total** | | **2.55** | **1.37** | **4.65** | **0.49** |

Note: 1 = not applicable at all; 2 = rarely applicable; 3 = moderately applicable; 4 = largely applicable; 5 = totally applicable.

On the other hand, in relation to the comments and feedback obtained from the participants in this study, the key findings of the study regarding the CollabVR virtual reality platform are summarized. These findings highlight both the positive features perceived by users and the distinctive differences of CollabVR compared to traditional online learning platforms. These tables present an analysis of the results from the responses obtained from participants based on the questionnaire about the perception of using CollabVR, highlighting the most beneficial and distinctive aspects they identified.

Table 2 illustrates the positive aspects of CollabVR, highlighting features that participants found particularly attractive and useful. These aspects include the ease of access and quality of personal interactions, the motivating environment provided by proximity chat, and the immersive experience that facilitates a deeper connection.

**Table 2.** Positive Aspects of CollabVR.

| Positive Aspects | Description |
|------------------|-------------|
| Ease of access and personal interaction. | Users appreciated the accessibility and the opportunity to interact more personally and directly, fostering meaningful relationships. |
| Proximity chat and motivating environment. | Highlighted for its functionality and the creation of a fun and engaging experience, essential for engagement in activities. |
| Immersive experience and deep connection. | Valued for its interactivity and ability to facilitate a deeper connection, increasing the feeling of being surrounded by peers. |

Table 3 presents the ways in which CollabVR differs from traditional online learning platforms. Highlights include improved interactivity and reduced distractions, immersive communication, and a sense of presence.

**Table 3.** Comparative advantages of CollabVR over blackboard.

| Characteristics | Description | Example |
| --- | --- | --- |
| Distraction avoidance | Perceived as a platform that minimizes common distractions, favoring instant and effective communication. | When establishing communication with other users, doubts or questions were answered instantly, which avoids situations such as unread messages or messages left "seen", a situation that often occurs through text communication on traditional platforms. |
| Immersive communication | Communication with movements and voice proximity allows for more immersive communication between users. | Immersive communication allows users to express themselves with hand and head gestures, improving eloquence. Voice proximity communication makes it easy to interact with multiple groups simultaneously and at any time, without interrupting others, something not possible on traditional platforms. |
| Sense of presence | Highlights the experience of being more present in the virtual environment, intensifying the feeling of connection and presence compared to traditional platforms. | Users experienced feeling "surrounded by people", something that is not experienced to such an extent on traditional platforms. |

## 6. Discussion, Limitations, and Future Work

In this study, CollabVR has been evaluated as a means to improve perceived social interaction in a remote educational environment, where this SVR platform was compared with Blackboard Collaborate, a traditional web platform. The total mean score of the sociability scale for the first group (Blackboard Collaborate) was $2.55 \pm 1.37$. This result aligns with the findings of Savci et al., where they reported a mean score of $2.35 \pm 0.75$ using Zoom as a collaborative web platform [32]. This suggests that, despite the socialization functionalities offered by traditional web platforms, levels of socialization remain moderately low. Furthermore, the consistency in these results suggests a possible trend toward limited socialization in online collaborative web environments, at least within the settings examined in the research. These parallels underscore the relevance of inquiry in CollabVR, aiming to address and potentially improve socialization in distance learning. These findings regarding the low level of socialization on web platforms should inspire more work in developing better features for them or the search for new technologies that do not replace, but rather complement, the current state of distance learning. Adopting existing VR applications, such as Mozilla Hubs, for the development of ECAs in education, is a good starting point. Although these applications are not specifically designed or optimized for management by universities, beginning with them is strategic. However, it is important to recognize that any process of technological adoption in education is inherently slow, requiring ongoing adaptations and technological advancements [33].

In contrast, the perceived level of socialization of the second group, which participated in ECAs in CollabVR, was significantly higher, with a total mean on the sociability scale of $4.65 \pm 0.49$. The results have shown that VR not only favors greater opportunities for socialization but also allows for more organic and spontaneous socialization opportunities during the development of activities. This represents a fundamental catalyst to strengthen personal relationships, allowing simultaneous socialization spaces and generating a greater sense of belonging, critical elements in building a sense of community among students.

On the other hand, one of the limitations was the representativeness of the users; although the study included a segment of the student population, it may not reflect the complete diversity of the global university population. This aspect is crucial since students' experiences and needs can vary widely in different cultural and educational contexts. Furthermore, the fact that the study was conducted in a specific context of students enrolled remotely, who lack in-person social interactions, raises questions about whether

the results would be replicable in a group of students with regular access to physical and social interactions.

Likewise, the availability of users' HMD peripherals is one of the main accessibility concerns. To address this issue, support has been implemented for "Desktop VR", which is a less immersive type of virtual reality; 3D virtual worlds like Second Life are examples of this [34]. Although the immersive experience is not the same, it is much cheaper and more accessible than the most immersive forms of virtual reality. However, it is important to highlight that technological advances in the field of artificial intelligence (AI) and trends in the technological advancement of virtual reality technologies are increasingly reducing this gap. For example, the integration of AI technologies that simulate hand and head movements through a Webcam allows users without HMD to access more functionality without the need for additional hardware. Additionally, advancement in VR technology is decreasing the costs and sizes of peripherals, making them more accessible to a wide range of users.

Initial research highlighted that SVR environments could offer a more suitable means of addressing these challenges compared to traditional virtual interaction methods. With this precedent, CollabVR was designed incorporating essential features that facilitate a rich and immersive socialization experience. Looking forward, it is vital to investigate the long-term impact of SRV use on students' social interaction and academic performance. It would be interesting to see how the continued integration of SVR platforms into the educational curriculum influences learning and the development of social skills. Finally, conducting comparative studies between different extended reality technologies could provide valuable insights to optimize social interaction in virtual environments, significantly contributing to the development of a more inclusive and effective pedagogy for the future of distance education. Nowadays, there is little use for VR technologies to promote socialization spaces related to university life. By providing support technologies that can complement the services provided by universities, these technologies can help many; the road is long but not impossible.

## 7. Conclusions

In this study, CollabVR, an SRV platform designed to improve perceived social interaction in remote educational environments was tested. The findings indicate that CollabVR has been effective in increasing social interaction among university students who participated in ECAs remotely, with an average of $4.65 \pm 0.49$ compared to traditional platforms with an average of $2.35 \pm 0.75$, according to the sociability scale.

Likewise, the results indicated that the main CollabVR functionalities compared to traditional web platforms were the feeling of presence, immersive communication, and the reduction of distractions, aspects that have greater influence in immersive VR environments. These findings validate the initial hypothesis and highlight the potential of virtual reality as a powerful tool to overcome the challenges inherent to socialization in virtual environments.

**Author Contributions:** Conceptualization, D.J. and B.M.; data curation, D.J. and B.M.; investigation, D.J. and B.M.; methodology, D.J. and B.M.; software, D.J. and B.M.; supervision, C.S.; validation, D.J., B.M. and C.S.; visualization, C.S.; writing—original draft preparation, D.J. and B.M.; writing—review and editing, D.J., B.M. and C.S. All authors have read and agreed to the published version of the manuscript.

**Funding:** This research received no external funding.

**Data Availability Statement:** The data presented in this study are available on request from the corresponding author due to privacy and ethical restrictions.

**Acknowledgments:** The authors of the paper wish to thank the teachers for their constructive comments and for the support that they offered.

**Conflicts of Interest:** The authors declare no conflicts of interest.

## Appendix A. Sociability Scale Result

The results of the sociability scale for both groups are presented below.

### Appendix A.1. AS-IS

Students who participated in online ECAs through traditional platforms presented a score of 2.33 (Table A1).

**Table A1.** Sociability scale items and means for ECAs using traditional online platforms.

| Code | Item | Mean |
|------|------|------|
| Q01 | This platform allowed me to easily contact my teammates. | 2.67 |
| Q02 | I did not feel alone on this platform. | 3.17 |
| Q03 | This platform allowed me to get a good impression of my teammates. | 2.92 |
| Q04 | This platform allowed for spontaneous informal conversations. | 1.92 |
| Q05 | This platform allowed us to develop as a well-performing team. | 2.50 |
| Q06 | This platform allowed me to develop good working relationships with my teammates. | 2.33 |
| Q07 | This platform allowed me to identify with the team. | 2.33 |
| Q08 | I felt comfortable with this platform. | 3.17 |
| Q09 | This platform allowed for conversations unrelated to the workshop task. | 2.25 |
| Q10 | This platform allowed me to establish close friendships with my teammates. | 2.25 |

### Appendix A.2. TO-BE

On the other hand, the students who participated in the ECA "Virtual Public Speaking" presented a general score of 4.7 (Table A2).

**Table A2.** Sociability scale items and means for ECAs using CollabVR.

| Code | Item | Mean |
|------|------|------|
| Q01 | This platform allowed me to easily contact my teammates. | 4.83 |
| Q02 | I did not feel alone on this platform. | 4.83 |
| Q03 | This platform allowed me to get a good impression of my teammates. | 4.67 |
| Q04 | This platform allowed for spontaneous informal conversations. | 4.92 |
| Q05 | This platform allowed us to develop as a well-performing team. | 4.50 |
| Q06 | This platform allowed me to develop good working relationships with my teammates. | 4.50 |
| Q07 | This platform allowed me to identify with the team. | 4.58 |
| Q08 | I felt comfortable with this platform. | 4.67 |
| Q09 | This platform allowed for conversations unrelated to the workshop task. | 4.83 |
| Q10 | This platform allowed me to establish close friendships with my teammates. | 4.67 |

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
