# Peer review of "CollabVR: VR Testing for Increasing Social Interaction between College Students"

_computers, doi:10.3390/computers13020040_

Round 1

Reviewer 1 Report

Comments and Suggestions for Authors

The paper proposes a Social Virtual Reality (SVR) platform called CollabVR, designed to improve social interaction among distance university students through Extracurricular Activities (ECAs). The platform leverages technologies such as microservices, Unity3D, Photon Unity Networking (PUN), Oculus Quest 2, and Amazon Web Services (AWS). CollabVR offers four virtual reality environments that foster socialization: Open Field, Presentation Room, Night Fire Pit, and Futuristic Room. The platform also includes moderation functionalities, activity creation tools, and activity metrics. This idea is quite interesting; however, I have few concerns which is given below.

1. Please elaborate few sentences in the abstract such as ". Leveraging 11 technologies such as Unity3D, Photon Unity Networking, Oculus Quest 2 and AWS, it aims to mit- 12 igate this social interaction deficit (DIS)."

2. Too many abbreviations make the article more ambiguous and hard to understand the specific idea. I will recommend to have a look at the following articles "Impact of stalling on QoE for 360-degree virtual reality videos", "Impact of the impairment in 360-degree videos on users VR involvement and machine learning-based QoE predictions". It will give good idea about the organization of work related to virtual reality.

3. The introduction section is lengthy. I will recommend to make it more concise. 

4. What model you have used for this study?

Reviewer 2 Report

Comments and Suggestions for Authors

Dear authors,

Thank you for the opportunity to review your manuscript. Here are some observations and suggestions that I hope help you in refining your manuscript:

* In the introduction you frame your context as a pandemic era setting where online learning had taken off since the pandemic. I think this kind of framing hides the fact that distance education, in the form of online asynchronous and synchronous learning, has existed for well over 30 years now in many contexts. There is plenty of of academic literature to actually disprove the issues like loneliness, depression, no socialization that "emerged" during the pandemic.  Much of that pandemic literature is problematic in nature in that it corelates these things with online learning but they weren't caused by online learning.

* What was the rationale for picking the Sociability Scale by Kreijns et al? How and why does it measure what you want to measure? For instance, why didn't you use another framework like Social Presence (part of the community of inquiry framework)?

* Your Sample Size seem rather small. Are the findings statistically significant?

* You mention in your design that the study took place in October. How many sessions did students undertake with each of these tools in the two groups? What it enough of an experience to really get to know what students perceive these technologies to be? 

* For your study population, what is their background? Are they familiar with both tools used (in both groups)? How might their previous experiences impact your study? Also, in the "as is" group, how much interaction was in the class?  Was the Blackboard Collaborate session essentially a broadcast? Or was it an interactive learning session? The design of how the technologies are used is an important factor.

* What does "remote" mean? I know that during COVID many universities didn't want to call online learning what it is, so they developed a euphemism for synchronous online learning.  I haven't seen the term used since we returned to "normal".  What does remote mean?  If it means synchronous online learning, then the term synchronous online learning or vILT (virtual instructor led training - common in enterprise/corporate settings) might be more appropriate. 

* small issue: On p. 9 - "The questionnaire design is based on the Sociability Scale, which measures perceived 252 sociability in computer-assisted collaborative learning environments [14]." - citation 14 doesn't seem to discuss this.  I think this is a typo.

* On p12 you indicate that there was an issue with obtaining HMDs for all participants, so what was used was a desktop application.  At that point, it's no longer virtual reality, but rather a kind of virtual world.  The heuristics and usability are different which might impact how your frame things.  What does comparable literature from virtual worlds research (e.g., second life) say?

Reviewer 3 Report

Comments and Suggestions for Authors

The article is an interesting demonstration of the use of the VR environment as a tool for initiating and supporting communication between students. Although his conclusions yield the expected results when comparing the use of LMS and VR and are not surprising, it is an interesting experiment.

Positives:

+ clearly and fluently written text

+ minimal deviation from the topic

+ interesting concept of the experiment

+ clear, albeit brief evidence of the difference between the perception of the web vs. VR environments investigated

+ understandable discussion

Comments:

- the goal of the article: the authors do not measure increased interaction, but increased PERCEPTION of interaction

- not very precisely implemented part Related work, maybe it would like to divide it into a part dedicated to technologies (where ColabVR is certainly not the only tool) and a part covering the contribution of the VR environment (which in principle is already included in the current text)

- at the beginning of the description, the origin of collabVR is not obvious, and this situation does not change until the end of the article

- it would be appropriate to describe what the respondents did during the individual stages of the experiment and what their role was, possibly also the time distribution of activities in VR.

- it is also not clear what the cooperation of the team consisted of, mentioned several times as the subject of the experiment

- when selecting respondents, their experience with the VR environment is not described, which can be a distorting factor of the results

- What is the source of Table 2 and 3? Is it just the assumption of the authors or are they the results of the questionnaire or its parts that are not described in the article?

- Discussion – „...should inspire more work in developing better features for them or the search for new technologies that do not replace, but rather complement, the current state of distance learning.“ - the authors could think about what might belong here, they could give some ideas.

- Next statement:The results have shown that the immersion offered by VR not only favors greater opportunities for socialization, but also allows for more organic and spontaneous socialization opportunities during the development of activities“, would need to be supported by evidence. Why do the authors think this is an immersion? Maybe it's the ability to use avatars and not show own face? The claim would like a link to resources or further discussion with participants...

- from 336 line probably 4xtypo SRV instead of SVR

I agree that VR appears to be a suitable tool for community building, it is questionable whether VR relationships will transfer to the real world, but that is not the topic of the current article.

I am glad that the authors are aware of the lack of respondents as a fundamental problem of the credibility of the research, but in most VR experiments it is so...

Overall, I consider the article to be a success, and after incorporating the mentioned comments, which are not of a fundamental nature, it will be possible to recommend the article for publication.

Round 2

Reviewer 1 Report

Comments and Suggestions for Authors

The authors disagreed with my comments and haven't addressed my concerns. In addition to my previous comments, I am highlighting few more issues. 

1. What is the role of the following metrics: "time during which a user is participating, the connection time of the users, and the number of users within the activities."

2. Why only these three metrics?

3. Dataset is described well. How "students participating in online ECAs" data have been collected.?

4. What was the methodology of data collection?

Comments on the Quality of English Language

Minor revision required

Round 3

Reviewer 1 Report

Comments and Suggestions for Authors

The authors have addressed all my concerns, and now the manuscript can accepted for publication.